

# How polar-to-midlatitude atmospheric teleconnections depend on regional sea ice fraction and global warming level

Carley E. Iles[1], Bjørn H. Samset[1], Marianne T. Lund[1]

[1]CICERO Center for International Climate Research, Oslo, Norway

*Correspondence to*: Carley E. Iles (carley.iles@cicero.oslo.no)

**Abstract**

The climates of the polar and mid-latitude regions are linked through teleconnections. The regional details of these relationships, and how they may change with global warming, are however still uncertain. Using two large ensembles of coupled climate model simulations (CESM2, ACCESS-ESM1.5) and a composite analysis, we investigate the statistical

relationships between sea ice variability and atmospheric circulation patterns, and how they evolve with sea ice retreat for both poles, including sensitivity to sea ice region in the Arctic. We find that relationships between sea ice amount and sea level pressure (SLP), the North Atlantic jet stream, and surface air temperature (SAT), depend on the region where sea ice varies. For instance, the North Atlantic jet shifts southwards when sea ice is low in the Labrador Sea, but shifts northwards and/or weakens for low Okhotsk sea ice and shifts northwards and strengthens for low Chukchi-Bering sea ice. We also investigate

the circulation patterns associated with changes in Antarctic sea ice. For the Arctic, circulation patterns tend to persist with global warming, until around 3 or 4 °C, when the ice edge has retreated substantially. In the Antarctic, patterns are sensitive to warming also at lower global warming levels for some seasons and variables, but are otherwise often persistent across warming levels. Lagged analysis suggests that the instantaneous relationships mostly reflect the atmospheric conditions contributing to low sea ice, with weaker or altered patterns when sea ice leads. Our results emphasize the importance of regional

heterogeneity, and on using large ensembles or other statistically rich datasets, for assessing influences of polar climate change on mid-latitude weather patterns today and in a warmer climate. The overall persistence of teleconnection patterns between sea ice change and atmospheric circulation with global warming is encouraging, as it indicates that the main conclusions from current literature will be applicable also in a future, warmer world with less sea ice.

**Short summary**

Polar sea ice changes and midlatitude weather affect each other, but how these teleconnections play out differ between the poles and between sea ice regions. Knowing how they interact is important for climate risk assessments, but few studies have investigated how the teleconnections evolve with global warming. Using large ensembles of climate model simulations, we find teleconnections patterns that differ between sea ice regions, but are quite robust to changes in global surface temperature.



## 1 Introduction

Arctic sea ice has declined rapidly since satellite measurements began in 1979, and is projected to keep declining over the coming decades, with the first practically ice free annual minimum likely to occur before 2050 in all emissions scenarios (Fox-Kemper et al., 2021; Notz and Community, 2020). Antarctic sea ice increased gradually between the late 1970s and 2014 (Comiso et al., 2017; Parkinson, 2019), but has declined rapidly in more recent years, with the four lowest annual minimum extents in the observational record occurring in 2022-2025 (NSIDC, 2025). It is projected to decline over the 21st Century in climate model projections (Roach et al., 2020). On top of these long-term trends there is also considerable year to year variability in sea ice amount (e.g. Hobbs et al., 2016; Olonscheck et al., 2019; Serreze and Meier, 2019).

The climate features of the polar regions, including the amount of sea ice, come about, in part, through a set of two-way teleconnections with mid-latitude weather. These connections involve atmospheric and oceanic circulations, and include, e.g., winds, storms and precipitation patterns, including advection of heat and moisture, which affect sea ice formation, melt or break-up and drift (Bonan et al., 2024; Dörr et al., 2021; Eayrs et al., 2019; Hobbs et al., 2016; Olonscheck et al., 2019; Schroeter et al., 2017; Serreze and Meier, 2019). On the other hand, sea ice loss-related changes in albedo and fluxes of heat and moisture can influence lower latitude dynamics. Possible effects of pan-Arctic sea ice decline identified in the literature include an expansion of geopotential height over the Arctic, weakening and equatorward shift of the jet stream and storm tracks, intensification of the Siberian high and Aleutian low, weakening of the polar vortex with a subsequent negative North Atlantic Oscillation (NAO) pattern and associated continental cooling, and an increased likelihood of slow moving amplified waves in the jet stream and associated blocking and extreme weather (e.g. Barnes and Screen, 2015; Cohen et al., 2020; Screen et al., 2018 and references in all). While there is a growing body of literature examining these effects, considerable diversity remains between studies on the mechanisms and details of responses (e.g. Cohen et al., 2020; Screen et al., 2018). Suggested sources for discrepancies include sensitivity to model biases and background state, including the phase of internal variability (e.g. Overland et al., 2016; Smith et al., 2017; Xu et al., 2024) representation of the stratosphere (Cohen et al., 2020; Sun et al., 2015), and differences in study methods such as whether or not atmosphere-only or coupled models are used (Ayres et al., 2022; Deser et al., 2015, 2016; Screen et al., 2018; Smith et al., 2017), the size of the ensembles (Blackport and Screen, 2021; Peings et al., 2021; Ye et al., 2024), and use of sea ice perturbation experiments versus composite or regression analysis applied to free-running model simulations (Blackport and Screen, 2021; Delhaye et al., 2024; Smith et al., 2017). Such variations in methodological approach makes it complicated to compare and synthesize across studies exploring different mechanisms, regions, or time periods using different models.

Some recent studies also suggest that the atmospheric circulation responses and associated temperature patterns and effects on the northern hemisphere polar jet stream depend on sub-regional patterns of sea ice loss (Cohen et al., 2020; Levine et al., 2021; McKenna et al., 2018; Pedersen et al., 2016; Screen, 2017; Sun et al., 2015; Xu et al., 2024). This is relevant both for interannual variations in sea ice, which tend to be spatially heterogeneous, and subsequent seasonal weather prediction, and in terms of uncertainty in where sea ice will decline most rapidly with further global warming (Screen, 2017). For instance, sea



ice loss in the Barents-Kara region has been found to cause a weakening of the polar stratospheric vortex, which can later
propagate towards the surface and cause a negative NAO or Arctic Oscillation (AO) later in the winter (Cohen et al., 2020;
Delhaye et al., 2024; McKenna et al., 2018; Screen, 2017; Sun et al., 2015), while sea ice decrease in the Okhotsk or Chukchi
seas has been shown to enhance the polar vortex (Kelleher and Screen, 2018; McKenna et al., 2018; Sun et al., 2015; Xu et
al., 2024). Importantly, these findings suggest discrepancies between studies of pan-arctic sea ice loss could be partly due to
differences in the spatial pattern of the sea ice forcing. With some exceptions, however, many studies have focused on only
one or a couple of regions.

Moreover, comparatively less research has been done on the effects of changes of Antarctic sea ice on the climate. Existing
modelling studies suggest a weakening and equatorward shift of the jet stream, equatorward shift in storm tracks, and a negative
Southern Annular Mode (SAM) response with reduced sea ice (Ayres et al., 2022; Ayres and Screen, 2019; Bader et al., 2013;
England et al., 2018), and the opposite for sea ice increase (Smith et al., 2017). Ayres and Screen, (2019) and England et al.,
(2018) also find a weakened Southern Hemisphere polar vortex with sea ice decrease, analogous to the effects Arctic sea ice
on the NH polar vortex. These jet stream and SAM responses are the opposite to that projected due to global warming (Ayres
and Screen, 2019; England et al., 2018), adding to the importance of improving understanding of teleconnections involving
sea ice loss in the Southern Hemisphere.

Here we perform a comprehensive analysis of the statistical relationships between sea ice variations and atmospheric
circulation, from year to year and under global warming, for both the Arctic and Antarctic. Furthermore, we systematically
characterize this relationship for both pan-Arctic sea ice variability, and for five sub-regions. The main objective is to quantify
the polar to midlatitude teleconnections as sea ice retreats, exploring the sensitivity to regional sea ice loss, and also how the
teleconnection patterns evolve with global warming and thus sea ice fraction. While a few studies have suggested that responses
to sea ice loss are not consistent for different amounts of overall sea ice loss, or for incremental changes from different starting
levels of sea ice (Chen et al., 2016; McKenna et al., 2018; Peings and Magnusdottir, 2014; Petoukhov and Semenov, 2010;
Semenov and Latif, 2015), the sensitivity of the atmospheric response to polar-wide and regional sea ice reduction has not yet
been explored over a large range of sea ice states in a consistent manner.

We use simulations of the historical period and future projections from two large ensembles of coupled climate model
simulations. Large ensembles are an information rich resource for this kind of question, as they contain a large sample size of
climate variability across various levels of background global warming, which has been shown to be necessary for examining
associations between sea ice loss and mid latitude climate, particularly for dynamics related variables (Blackport and Screen,
2021; Peings et al., 2021; Ye et al., 2024). The use of coupled model simulations is important as it allows for an exploration
of the full range of interactions between sea ice, the atmosphere, and the ocean (Ayres et al., 2022; Deser et al., 2015, 2016;
England et al., 2020b, a). We use a composite method to look at the atmospheric circulation patterns associated with sea ice
variations, similar to a method recently applied to preindustrial control simulations (Delhaye et al., 2024), but applied also to
examine the linkages under future conditions. Specifically, we examine the relationship of surface air temperature (SAT), sea
level pressure (SLP) and the North Atlantic and Southern Hemisphere polar jet streams to sea ice variations, the latter included



since they are a key feature affecting mid-latitude weather. Because this method may capture both the influence of the atmosphere on the sea ice, and the effect of sea ice on the atmosphere, we also perform a lagged analysis in an attempt at 100 disentangling the two.

The remainder of this paper is organised as follows. In section 2 we outline the data and methods used, while section 3 presents results for the Arctic, starting with Pan-Arctic sea ice changes (section 3.1), and then regional sea ice variations (section 3.2). Section 3.3 presents results of the lagged analysis and section 3.4 examines linkages between the North Atlantic jet stream and regional Arctic sea ice variations. Finally, section 4 focuses on the corresponding Antarctic analysis while results are discussed 105 and summarized in sections 5 and 6.

## 2 Data and Methods

### 2.1 Data

We conduct the analysis using existing large ensembles of CMIP6-generation coupled climate simulations for both the historical period and future projections. This allows a large sample of sea ice variability and associated atmospheric circulation 110 states for various levels of background warming to be examined. We use large ensembles from two models: CESM2 which has 100 members for the historical period (1850-2014) and the future under assumptions of relatively high emissions (SSP3-7.0; 2015-2100) (Rodgers et al., 2021), and ACCESS-ESM1-5 which has 40 members for the historical period, and projections for 4 different SSPs (SSP1-2.6, 2-4.5, 3-7.0 and 5-8.5)(Mackallah et al., 2022). Both models perform well in terms of total Arctic sea ice area in March, although CESM2 has a low bias in September (Notz and Community, 2020; Shu et al., 2020; 115 Watts et al., 2021). Both models have a very good representation of total Antarctic sea ice area and interannual variability during both the sea ice maximum and minimum (Roach et al., 2020; Shu et al., 2020). We calculate total sea ice area over a region by multiplying the sea ice fraction (or concentration / 100) by the area of the corresponding grid cell, and then calculate the sum over the region in question. To examine the associated temperature and atmospheric circulation patterns, we use monthly surface air temperature, sea level pressure and zonal wind at 250 hPa or 700 hPa to look at changes in the Northern 120 Hemisphere and Southern Hemisphere polar jet streams respectively. For CESM2 we use the closest average pressure level, while for ACCESS-ESM1-5 precise pressure levels are available.

### 2.2 Methods

For a given large ensemble and region, we select years with low or high sea ice area for a given amount of global warming and then examine the difference in SLP, SAT and 250 or 700 hPa zonal wind between the two sea ice states using a composite 125 analysis (i.e. looking at the average SLP etc from a composite of all years with high sea ice amounts, compared to one consisting of years with low sea ice amounts). These differences in atmospheric variables thus reflect the difference associated with internal variability of sea ice at different levels of background warming, rather than with a change of sea ice from pre-industrial levels to the warming level in question. We select years with sea ice area under the $30^{\text{th}}$ percentile for a given





warming level as "low sea ice" and years over the 70th percentile as "high sea ice" as illustrated in Fig. 1b and c. We then take the mean of the SLP, zonal wind or SAT anomalies (calculated with respect to 1850-1900) across all high or low sea ice years for a given warming level (see example in Fig. 1 d,e). Finally, we take the difference between these two composites (low minus high) as the atmospheric conditions associated with low sea ice anomalies (e.g. Fig. 1f). We test the statistical significance of this difference using a two sample, two tailed t-test at the 95% significance level. If the 30th percentile of sea ice reaches zero then results are not shown, although this does not actually happen for the months presented here. Concerning the background levels of global warming, we collect together all years from all ensemble members with annual mean global surface air temperature within +/- 0.25 °C of the warming level in question, and then look at high vs low sea ice years within each warming level (Fig. 1b, c). We show results for 0, 1, 2, 3 and 4°C relative to the preindustrial (1850-1900). We calculate results for pan-Arctic sea ice (i.e. all sea ice in the Northern Hemisphere (NH)), pan-Antarctic sea ice (likewise but for the Southern Hemisphere (SH)) and for a number of Arctic subregions, which are defined following Delhaye et al., (2024) and are shown in Fig. 1a and Table 1. This method has the advantage that the sea ice changes are realistic (to the extent that the models simulate it well) and internally consistent with the rest of the climate state, as opposed to imposing a fairly arbitrary sea ice perturbation as was done in some earlier studies. The full set of scatterplots for all regions and both models can be found in Fig. S1 and S2.

Table 2 shows the number of years counted as high and low sea ice states for each warming level and each model. The sample size is very high, particularly at 0°C, when the climate was fairly stable at pre-industrial levels, and decreases with increasing global warming levels. Results at higher warming levels may consequently be less robust than those at lower ones, but these are nevertheless still large sample sizes and comparable to the ensemble sizes used in the Polar Amplification Model Intercomparison Project (PAMIP), which requires a minimum 100 members (Smith et al., 2019).

Simultaneous anomalies in sea ice and circulation variables or SAT could reflect the response of the atmosphere to sea ice reduction, or the atmospheric conditions that lead to sea ice reduction, or a mixture of the two. While our study and approach are not designed with the primary goal of quantifying cause and effect, we perform a lagged analysis, looking at atmospheric anomalies in the months both before and after the sea ice changes to discuss and provide first order insights into what may be cause and what may be effect.





**Figure 1: a)** The Arctic sea ice regions considered (see also Table 1) **b)** Interannual variability in January sea ice area for the Barents-Kara Seas vs annual mean global surface air temperature (GSAT, °C) for the CESM2 large ensemble for 1850-2100. Red boxes illustrate the selection of points for "high" (>70th percentile) and "low" (<30th percentile) sea ice areas for different global warming levels. **c)** as for panel b) but for January pan-Arctic sea area ice for ACCESS-ESM1-5 and the inclusion of multiple scenarios. Panels **d)** and **e)** show the average sea level pressure anomalies relative to the preindustrial period (1850-1900) associated with low and high January pan-Arctic sea ice states respectively at the example warming level of 0°C for ACCESS-ESM1.5. Panel **f)** shows the difference between panels d) and e), and thus the average difference in SLP for winters with low sea ice minus those with high sea ice. Stippling in f) indicates a significant difference of the mean of SLP between low and high sea ice states using a t-test (p<0.05). Contours in d-e) indicate climatological mean SLP over the preindustrial period (1850-1900), with a contour interval of 4 hPa, with contours 1014 hPa or less dashed.



| Region | Latitude | Longitude |
|---|---|---|
| Pan-Arctic/ Northern Hemisphere (NH) | 0–90 °N | -180–180°E |
| Barents-Kara Seas | 70–82 °N | 15–100 °E |
| Sea of Okhotsk | 40–63 °N | 135–165 °E |
| Chukchi-Bering Seas | 50–82 °N | 170–160 °W |
| Labrador Sea | 55–80 °N | 70–40 °W |
| Greenland Sea | 50–75 °N | 40–15 °E |
| Antarctic | -90–0 °N | -180–180 °E |

**Table 1: The sea-ice regions used in this analysis, following** (Delhaye et al., 2024)**.**


| Warming level | Number of points selected for high and low sea ice states | |
|---|---|---|
| | CESM2 | ACCESS-ESM1.5 |
| 0 °C | 3397 | 1557 |
| 1 °C | 588 | 487 |
| 2 °C | 381 | 1171 |
| 3 °C | 342 | 546 |
| 4 °C | 288 | 224 |

**Table 2: Number of years selected as having high (>70[th] percentile) and low (<30[th] percentile) sea ice per global warming level and model. Note that these numbers are the same for every region and that there are the same number of cases for both high and low sea ice states.**

## 3 Results for Arctic sea ice variability

We first examine the relationships between Arctic sea ice variability and atmospheric circulation. We focus on winter as this is i) when the strongest atmospheric circulation patterns associated with Arctic sea ice variability were found (results for summer and autumn were much weaker, ii) when the stratospheric pathway leading to a delayed negative NAO response to Barents-Kara sea ice reduction is potentially active, and iii) the focus of many studies with which we compare our findings (Section 5). We focus specifically on January sea ice changes. We first examine instantaneous relationships between sea ice and atmospheric circulation, and then examine associations when sea ice or the atmosphere lead or lag by a couple of months, to better understand what is cause and what is effect.





### 3.1 Pan-Arctic sea ice variability and atmospheric linkages

We first focus on the relationships between pan-Arctic sea ice variability and atmospheric circulation. Figure 2 shows how sea
ice fraction, surface air temperature and sea level pressure differ between winters with anomalously low and high total area of
Arctic sea ice at various levels of background global warming for CESM2. Figure 2a shows the difference in sea ice
concentration between low and high sea ice states for each global warming level. It can be seen that the regions where sea ice
varies the most from year to year move as sea ice retreats with global warming. To begin with the biggest variations are found
in the Barents-Kara and Greenland Seas. At 3 or 4°C of global warming the Chukchi and Beaufort seas also undergo large
interannual variability in sea ice fraction, whilst the region of maximum variability for the Barents-kara seas moves eastwards.
Thus, the region of forcing from sea ice loss to the atmosphere changes through time.

Figure 2b shows the associated SAT patterns for low minus high sea ice states for pan-Arctic sea ice variability. The Arctic as
a whole is warmer when pan-Arctic sea ice is anomalously low, and the regions with the largest warm anomalies are situated
above and in the vicinity of the areas of the largest low sea ice anomalies, and thus also move towards the pole as global
warming progresses. Low pan-Arctic sea ice is also associated with areas of significant cool anomalies over the midlatitude
continents, including North America and mid latitude Eurasia at lower warming levels, and just the latter at higher warming
levels. Findings for ACCESS-ESM1-5 are similar (Fig. S3), although it starts with more sea ice overall, and ends with less at
4°C (Fig. S1a,b). The exact regions where sea ice varies the most thus differ a little from CESM2, e.g. additionally including
the Labrador sea at 0°C and including a larger part of the central Arctic ocean at 4°C, with corresponding effects on the location
of the warmest temperatures associated with low sea ice years (Fig. S3d,e). ACCESS-ESM1-5 does not show cooling over
North America when sea ice is low but agrees with CESM2 about cooling over the Eurasian mid latitudes (Fig. S3b).

Figure 2c and d show the SLP patterns for low and high sea ice states respectively as anomalies relative to the preindustrial
period. They thus primarily show the global warming signal in SLP patterns, which consists of reduced SLP over much of the
Arctic with high pressure anomalies further south, resembling a positive Arctic Oscillation (AO) pattern. ACCESS-ESM1-5
shows broadly similar patterns, but with weaker and less widespread positive anomalies (Fig. S3c,d). Figure 2e shows the
difference between these top two rows for CESM2, and thus the SLP pattern associated with low sea ice states compared to
high sea ice states at different levels of background warming. Low sea ice area is associated with significant high-pressure
anomalies over northern Eurasia, a weakened Aleutian Low and low pressure elsewhere. These patterns largely persist through
the different background global warming levels, but with a switch in sign of the anomalies over the Aleutian Low at warming
levels of 3°C or higher. This northern Eurasian high pressure anomaly with low pan-Arctic sea ice is also found in ACCESS-
ESM1.5, but with low pressure across most of the rest of the Northern Hemisphere and more persistence of the pattern into
higher warming levels (Fig. S3e).



**Figure 2: Row a) shows the average difference in January sea ice fraction between winters with anomalously low and high total Arctic sea ice area at a range of background global warming levels (ranging from 0 to 4°C; columns) for CESM2. Row b) is like row a) but showing the difference in January surface air temperature between low and high sea ice winters. Rows c) and d) show the average SLP anomalies with respect to the preindustrial period (1850-1900) for low and high sea ice winters respectively at the various levels of background global warming. Row e) is like rows a) and b) but showing the difference in SLP between low and high sea ice winters, (i.e. the difference between rows c) and d)). Stippling indicates a significant difference of the mean of the given variable between low and high sea ice states using a t-test (p<0.05). Contours in rows c-e) show climatological mean SLP over the preindustrial period (1850-1900), with a contour interval of 4 hPa, with contours 1014 hPa or less dashed. Numbers in the top right corners of row a) indicate the change in sea ice area between low and high sea ice states in units of a million $km^2$.**



## 3.2 Regional Arctic sea ice variations and atmospheric linkages

Previous research suggests both that sea ice decline in different regions can have differing effects on the atmospheric
circulation, including possible cancellation or non-linear interactions, but also that different atmospheric circulation patterns
can cause sea ice decline or increases in different regions. We therefore examine the atmospheric circulation patterns that are
associated with regional sea ice variations.

Figure 3 shows the difference in sea ice fraction and SAT between low and high sea ice states for various regions at various
levels of global warming. Note that the regional sea ice variations are larger than the pan-arctic ones for a given region because
different regions may have high or low sea ice in a given year, which tends to cancel out and lead to muted variations in the
pan-arctic case. As in the pan-arctic case, the strongest sea ice anomalies are near the ice edges, and they move as global
warming increases and the sea ice edge retreats poleward. Although sea ice anomalies sometimes extend a little outside the
selected region, there are mostly only weak (albeit significant due to our very large sample size) correlations of sea ice
variations between regions (Table S1). The largest correlation of r=0.32 is between sea ice area in the Labrador and Greenland
seas in CESM2. This implies that, for the most part, our regional results should not be affected much by sea ice variations
outside the region of interest.

SAT is again anomalously high over the region of low sea ice anomalies and neighbouring regions, whilst a significant remote
cooling is seen elsewhere, the location of which depends on where the sea ice varies. For instance, there is midlatitude Eurasian
cooling when sea ice is low in the Barents- and Kara seas, cooling over North America for low sea ice in the Okhotsk and
Chukchi-Bering seas, and high-to-mid-latitude Asian cooling for low sea ice conditions in the Labrador and Greenland seas.
These SAT patterns likely contain a combination of direct heating from sea ice reduction, due to increased heat flux from
ocean into atmosphere, and the effect of both the atmospheric circulation patterns that lead to low sea ice conditions, and those
arising from the low sea ice state (McKenna et al., 2018; Sun et al., 2015) (see also Section 3.3). The pattern stays broadly
similar with background global warming level, but does evolve a little, generally with less wide-spread cool anomalies during
low sea ice years relative to high sea ice years as global warming increases, whilst the area of strongest warm anomalies
follows the retreat of sea ice. Results are broadly similar for ACCESS-ESM1.5 (Fig. S4).





**Figure 3: Left side-** mean difference in sea ice fraction for January for low minus high sea ice states for different regions (NH - Northern Hemisphere and selected Arctic seas) and selected global warming levels (ranging from 0 to 4°C) for CESM2. The black outlines show the sea ice regions examined. **Right side-** the same but for surface air temperature. Stippling indicates a significant difference of the mean of sea ice fraction or temperature between low and high sea ice states using a t-test (p<0.05). Numbers in the top right corners indicate the change in sea ice area between low and high sea ice states in units of a million km$^2$.





**Figure 4: The difference in SLP for January for low minus high sea ice states in different regions (NH - Northern Hemisphere, and selected Arctic seas) and selected levels of global warming (ranging from 0 to 4°C) for CESM2. The region where sea ice is examined is shown with a black outline. Stippling indicates a significant difference of the mean of SLP between low and high sea ice states using a t-test (p<0.05). Contours indicate climatological mean SLP over the preindustrial period (1850-1900), with a contour interval of 4 hPa, while contours 1014 hPa or less are dashed. Numbers in the top right corners indicate the change in sea ice area between low and high sea ice states in units of a million km².**



Figure 4 shows the SLP patterns associated with low minus high sea ice anomalies for various Arctic sea ice regions and different levels of background global warming. The SLP patterns associated with sea ice variations in the different regions are very different. Low sea ice in the Barents-Kara Seas is associated with a pattern somewhat like a positive NAO, but with an additional strong high pressure anomaly over northern Eurasia. For the Okhotsk Sea, low sea ice is associated with a weakened or southward shifted Aleutian Low, whilst for the Chukchi and Bering seas there is a pressure dipole with low pressure to the

west and high pressure to the east of the area of reduced sea ice. Anomalously low sea ice in the Labrador Sea is associated with a negative AO pattern. Finally, low sea ice in the Greenland Sea is associated with co-located low pressure with high pressure to the east over northern Eurasia. These patterns tend to persist with increasing global warming levels, although sometimes with some changes at the highest warming levels (e.g. for the Barents and Kara seas and for the Chukchi and Bering seas). This could be due to the migration through time of where sea ice varies year to year as the ice edge retreats. The SLP

patterns found in ACCESS-ESM1-5 are similar but somewhat weaker and a bit less widespread for some regions e.g. for low Labrador sea ice (Fig. S5). These SLP patterns would generally tend to advect warm air into the region of low sea ice, and cold air into the cold regions in Fig. 3.

### 3.3 Lagged analysis for regional Arctic sea ice changes

In the above analysis it is not obvious whether the SLP and SAT patterns are the result of low sea ice, or contribute to low sea

ice, or a mixture of the two. To explore cause and effect we use a lagged analysis in which we look at the SAT and SLP patterns in the months leading up to and following low sea ice in January. This allows us to get a first order idea of how sea ice variability can affect weather and climate outside of the Arctic, and how these effects might change as global warming progresses. Figure 5 shows the SAT patterns that lag regional sea ice variability, from two months before (lag -2 and -1) and two months after (lags +1 and +2) the Januarys with anomalously low sea ice. This is shown for the example global warming

level of 2 °C. This level was chosen as it is in the middle of the range of warming levels considered, and the results are also fairly representative of the other warming levels. The temperature patterns one month before low sea ice (lag -1) are very similar to the instantaneous patterns (lag 0), and even at lag -2 there is a similar but slightly weaker version of the same pattern. This pattern then fades away over the positive lags, with a weaker pattern in lag +1 than in lag -1. A similar evolution can be seen for ACCESS-ESM1.5 (Fig. S8). The similarity of the results for lag -1, -2 and 0 suggests that lag 0 is primarily reflecting

the temperature patterns that cause a decrease in sea ice in the given region. However, sea ice exhibits a high degree of autocorrelation from month to month, and there remains a separation of high and low sea ice states for a few lags before and after lag 0 (Fig. S6 and S7). Therefore, any response to reduced sea ice could also appear in the lag -1 and -2 plots, since sea ice is already low. Notably, even in the Okhotsk Sea, where there is no sea ice in November (i.e. lag -2, Fig. S6), we still see a consistent temperature evolution across negative lags as in the other regions. This may lend support to the idea that these are

the SAT patterns driving a reduction in sea ice, or it could be that the sea surface temperatures preceding low sea ice are warmer. The lag +1 pattern is somewhat weaker than that at lag -1 and 0, but the strongest warm anomalies correspond very





closely to the pattern of sea ice reduction, suggesting that this aspect could be a response to sea ice forcing. Likely cause and effect are intermingled in this analysis and are not easy to separate cleanly with a composite method.

**Lagged temperature for low minus high January sea ice at 2°C, CESM2**



**Figure 5: Surface air temperature patterns associated with low minus high sea ice in various regions (NH - Northern Hemisphere, and selected Arctic seas) at various lags in CESM2 at 2°C of global warming. Negative lags represent the temperature pattern 1-2 months before low January sea ice, and positive lags, the temperature 1-2 months afterwards. Stippling indicates a significant mean temperature difference between low and high sea ice states using a t-test (p<0.05).**

We also plot lagged patterns for SLP for lag -1 to lag +1 (Fig. 6), while lag -2 and +2 are shown in Fig. S9. Since the SLP patterns at positive lags (sea ice leads SLP) are less consistent across warming levels compared to SAT, we show results for a range of warming levels. SLP results for lag -1 were more consistent across warming levels, and are very similar to the results for lag 0 above, so we only show results for 2°C as an example.

For some regions e.g. the Labrador, Greenland and Okhotsk seas, the same pressure pattern at lag 0 largely persists at lag +1,

albeit weaker. For others, e.g. the Barents-Kara Seas, the pattern changes considerably at lag +1 – there are some differences between global warming levels, but at 0, 1, 3 and 4°C there is a pattern resembling a negative NAO/AO. A negative AO pattern is also seen at lag +1 for pan-Arctic (labelled NH in Figure 6) sea ice reduction for 0-2°C. Other regions retain some aspects of the lag 0 pattern at lag +1, but it evolves elsewhere. For example, for sea ice reduction in the Chukchi-Bering Sea, the high pressure anomaly to the East of the region disappears at lag +1, but other aspects persist, including low pressure over the

Aleutian low and high pressure over northern Eurasia. Broadly similar patterns persist at lag +2 (Fig. S9). Whilst there are some differences in patterns between warming levels, it seems likely that some of this is climate variability related, given the lack of clear progression across warming levels e.g. for the Barents-Kara case. This is despite our large sample size (see also (Peings et al., 2021)). In contrast to lag +1, lag -1 and even lag -2 (SLP leads sea ice) show very similar patterns to those at lag 0 (Fig. 6 and S9), suggesting that lag 0 is likely primarily reflecting the SLP patterns that contribute to low sea ice in a given

region.

For ACCESS-ESM1-5, SLP patterns at lag +1 are weaker and less distinct (Fig. S10). They include some aspects of similarity with CESM2 e.g. a similar but very weak overall pattern for low Okhotsk sea ice decline, a deepening of the Aleutian low for Chukchi sea ice decline, and a negative NAO (albeit weak and possibly clearer at lag +2 (Fig. S11), and much clearer when using 500 hPa geopotential height (Fig. S12)) for pan-Arctic and Barents-Kara sea ice reduction. For the Labrador and

Greenland seas, ACCESS-ESM1.5 shows more localised patterns at lag +1 with a relatively strong low pressure anomaly over the region of sea ice retreat that does not exist in the CESM2 patterns (Fig. S10). Again, some patterns at lag +1 retain some of the pattern at lag 0, but are weaker overall (e.g. for the Labrador, Greenland and Okhotsk seas). Once more, SLP patterns at lag -1, -2 and 0 are similar to each other, and also similar to those found in CESM2.

.






**Figure 6:** Sea level pressure responses in February to January variations in sea ice extent (i.e. lag +1) for different regions (NH - Northern Hemisphere, and selected Arctic seas) and selected global warming levels (ranging from 0 to 4 °C) for CESM2. Lag 0 (instantaneous relationship) and -1 (atmosphere leading sea ice) at 2°C are shown for comparison in the second and first columns respectively. Stippling indicates a significant difference of the mean of SLP between low and high sea ice states using a t-test ($p<0.05$). Contours indicate climatological mean SLP over the preindustrial period (1850-1900), with a contour interval of 4 hPa, while contours 1014 hPa or less are dashed. Numbers in the top right corners indicate the change in sea ice area between low and high sea ice states in January in units of a million km$^2$.





## 3.4 North Atlantic Jet and regional Arctic sea ice variations

We also examine the relationship between sea ice variability and the North Atlantic jet stream at various levels of background warming. The North Atlantic jet stream is an important driver of weather variability in North America and Europe. In Figure 7 we show anomalies of the zonal wind at 250 hPa across the North Atlantic for low minus high January sea ice states at lag 0 for the various sea-ice-regions and a selection of warming levels for CESM2. As can be seen in Fig. 7, the instantaneous relationship between January sea ice amount and zonal wind at 250 hPa depends on the region in which the sea ice varies. The relationships are similar between CESM2 and ACCESS-ESM1.5 for most regions except the Northern Hemisphere as a whole and the Barents-Kara seas (Fig. 7 and Fig. S13). When sea ice is anomalously low in the Okhotsk Sea, there is a northward shift and/or slowdown of the Atlantic jet across all warming levels. For low sea ice in the Chukchi- Bering Sea the North Atlantic jet stream strengthens and shifts northwards. Low sea ice in the Labrador Sea is associated with strengthened winds on the southern part of the jet, and reduced winds in the northern part, which could suggest a southward shift. Low sea ice in the Greenland sea is associated with strengthened wind on the northern side of the jet, and weakened to the south and North-East, which could imply a northward shift or a more zonal tilt. Similar patterns are largely found across all levels of background global warming, although they can differ a bit, particularly at the higher warming levels. For the Barents-Kara Sea, low sea ice is associated with a northward shift of the jet at low warming levels in CESM2 which evolves at higher warming levels, whilst in ACCESS-ESM1-5 relationships between Barents-Kara sea ice and zonal wind are generally weak (Fig. S13). The two models disagree about the relationship of the jet to Pan-Arctic (labelled NH in Figure 7) sea ice variability: CESM2 shows a northward shift at the western end of the jet and a southward shift on the eastern end when Arctic sea ice is low. ACCESS-ESM1.5 only shows the latter. These patterns evolve somewhat with increased global warming level.

Figure S14 compares the patterns at lag 0 with those at lag -1 and lag +1 for CESM2. Due to space constraints we only show the patterns at 2 °C for lags -1 and 0. For lag +1 we show results at 0, 2 and 4 °C. The zonal wind patterns at lag -1 (atmosphere leading sea ice) are generally quite similar to those at lag 0. For the Okhotsk sea, patterns at lag +1 are quite strong, and similar to those at lag 0. For the Labrador sea, patterns at lag +1 are a weakened version of those at lag 0. In contrast, for the Chukchi-Bering, Greenland and Barents-Kara Seas and Pan-Arctic sea ice reduction, the zonal wind patterns are often weak at lag +1 and not that consistent across warming levels. In ACCESS-ESM1-5, lags -1 and 0 are also generally similar to each other, and are often a weaker version of those in CESM2 (Fig. S15). However, the patterns in ACCESS at lag +1 are generally weak and indistinct, although they tend to be stronger at 4°C. The main exception is for the Labrador sea where a weak version of the lag 0 and -1 pattern remains at lag +1, similar to that in CESM2.





**Figure 7: Mean difference in January zonal wind at 250 hPa for low minus high January sea ice states in a selection of Arctic seas for a selection of global warming levels (ranging from 0 to 4 °C) in CESM2. Positive values denote an increase in wind blowing from west to east (or a slowdown from East to West). Also shown with contours is the climatological mean zonal wind over the preindustrial period, which indicates the average position of the North Atlantic jet stream (contour interval 5 m s⁻¹, negative contours are dashed). Stippling indicates a significant zonal wind difference between low and high sea ice states using a t-test (p<0.05). Numbers in the top right corners indicate the change in sea ice area between low and high sea ice states in units of a million km².**



## 4 The Antarctic

Finally, we turn to the Southern Hemisphere, focusing on pan-Antarctic sea ice variability in four months spread across the year: austral summer (January) and winter (July) and annual sea-ice maxima (September) and minima (March). Whilst total sea ice area is generally slightly lower in February (Fig. S6 and S7), it is almost as low in March, allowing more of a spread
between the months examined.

### 4.1 Antarctic sea ice and atmospheric linkages

First, we look at the associations between Antarctic sea ice variability, SLP and SAT at various levels of background global warming. Fig. 8 shows the difference in sea ice fraction between low and high sea ice states, and the associated difference in SAT for CESM2 for various months. It can be seen that the areas of maximum sea ice variability contract around Antarctica
as global warming continues and the ice edge retreats. In years where sea ice is anomalously low, temperature tends to be high, particularly over the regions with low sea ice, but much of the Southern Hemisphere also shows small, but statistically significant, warm SAT anomalies, with some smaller regions of statistically significant cool anomalies e.g. to the west of Antarctica in January and March. The regions of strongest warm SAT anomalies in low sea ice years migrate polewards in accordance with where sea ice varies the most as global warming progresses. These warm anomalies are weakest in summer
(January), which might be because the ocean and atmosphere have similar temperatures at this time of year, limiting the increase in heat flux when sea ice is removed (Ayres and Screen, 2019; Menéndez et al., 1999; Screen, 2017). Results are similar in ACCESS-ESM1-5 (Fig. S16), albeit with more extensive areas with significant cool SAT anomalies when sea ice is low, and a tendency for stronger warm anomalies over the Antarctic interior at higher warming levels in July and September.

Figure 9 shows the SLP patterns associated with anomalously low sea ice in the Antarctic in CESM2 at a selection of warming levels, and also for an example warming level (2°C) for lag -1 and +1. ACCESS-ESM1.5 is show in Fig. S17. Whilst in ACCESS-ESM1.5 these SLP patterns tend to be zonally symmetrical, for CESM2 they exhibit a fair degree of zonal asymmetry. There tends to be high pressure anomalies over the pole when Antarctic sea ice is low in most seasons in CESM2, particularly in March and July (austral autumn and winter), with low pressure around it. This high pressure also spills over
into the African and Australian sectors of the southern Ocean in March and January, and more towards the southern Pacific in July and September. There is some degree of persistence across global warming levels, but also some differences, particularly at 0°C in September and July, with a reversed sign over the pole for the former, with very weak patterns for the latter. Lag -1 tends to show a stronger version of the lag 0 patterns, whilst lag +1 retains some resemblance to lag 0.  In ACCESS-ESM1.5 we find a positive Southern Annular Mode (SAM) pattern in austral summer (January) when sea ice is low, and a negative
SAM pattern in the other seasons, particularly in austral winter and spring (July and September). This pattern strengthens with warming level in September, but is otherwise fairly consistent across warming levels in other months (except at 4°C in March). Lag -1 shows a stronger version of the lag 0 pattern, and lag -1 a weaker version.





**Figure 8: Left side- difference in sea ice fraction between low and high Antarctic sea ice states for selected global warming levels (ranging from 0 to 4°C) and months (January, March, July, September). Right side- the same but for surface air temperature. Stippling denotes significance with a t-test (p>0.05). Numbers in the top right corners indicate the change in sea ice area between low and high sea ice states in units of a million km².**







**Figure 9: The difference in sea level pressure for low minus high sea ice states for the Antarctic for selected levels of global warming (ranging from 0 to 4°C) and months (January, March, July, September) for CESM2. Middle three columns are for instantaneous relationships, i.e. lag 0, whilst lag -1 (atmosphere leads sea ice by one month; left column) and lag +1 (sea ice leads atmosphere by one month; right column) are shown for an example warming level (2°C) for comparison. Stippling indicates a significant difference of the mean of SLP between high and low sea ice states using a t-test (p<0.05). Contours indicate climatological mean SLP over the preindustrial period (1850-1900), with a contour interval of 4 hPa, while contours 1014 hPa or less are dashed. Numbers in the top right corners indicate the change in sea ice area between low and high sea ice states in units of a million km$^2$.**





## 4.2 Southern Hemisphere polar jet and Antarctic sea ice

**U700 winds for low minus high Antarctic sea ice, CESM2**

*[Figure: Polar projection maps arranged in a grid. Columns labeled Lag -1 (2°C), Lag 0 (0°C, 2°C, 4°C), Lag +1 (2°C). Rows labeled Jan, March, July, Sept.]*

Numbers in top-right corners:

| | Lag -1 (2°C) | Lag 0 (0°C) | Lag 0 (2°C) | Lag 0 (4°C) | Lag +1 (2°C) |
|---|---|---|---|---|---|
| Jan | 0.80 | 1.18 | 0.80 | 0.48 | 0.80 |
| March | 0.62 | 1.41 | 0.62 | 0.21 | 0.62 |
| July | 1.06 | 1.08 | 1.06 | 1.09 | 1.06 |
| Sept | 0.90 | 1.01 | 0.90 | 1.07 | 0.90 |

**Figure 10: The difference in zonal wind at 700 hPa (U700) for low minus high sea ice states for the Antarctic for selected levels of global warming (ranging from 0 to 4°C) and months (January, March, July, September) for CESM2. Middle columns are for instantaneous relationships, i.e. lag 0, whilst lag -1 (atmosphere leads sea ice by one month; left column) and lag +1 (sea ice leads atmosphere by one month; right column) are shown for an example warming level (2°C) for comparison. Stippling indicates a significant difference of the mean of SLP between high and low sea ice states using a t-test (p<0.05). Contours indicate the climatological mean U700 over the preindustrial period (1850-1900), showing only values over 10 ms⁻¹ to highlight the position of the jet stream, with a contour interval of 2.5 m s⁻¹. Numbers in the top right corners indicate the change in sea ice area between low and high sea ice states in units of a million km².**



Finally, we examine the relationship between the southern hemisphere polar jet stream and pan-Antarctic sea ice variability at different levels of background warming at different times of year. Figure 10 shows the patterns of 700 hPa zonal wind in the southern hemisphere associated with low Antarctic sea ice anomalies for CESM2. The equivalent for ACCESS-ESM1.5 is shown in Fig. S18. We use a lower atmospheric level than for the Arctic in order to pick up the polar jet rather than the subtropical jet. In this case, there are marked differences between the two models. For CESM2 the U700 wind patterns associated with low Antarctic sea ice are largely zonally asymmetrical. In January and March there is a strengthening and a weak equatorward shift of the jet over the Eastern South Pacific when sea ice is low, with a slowdown of the jet elsewhere. This kind of pattern is found over all three lags (-1 to +1). In July there is an equatorward shift of the jet at lag -1. This pattern weakens at lag 0 and +1 and becomes more of a slowdown of the jet over many regions, with a speed-up near South America. Patterns are somewhat similar in September to July. There are some differences across warming levels with the patterns at 0°C often being weaker and sometimes also different than at other warming levels for all lags (only shown for lag 0 here).

In ACCESS-ESM1.5 the U700 wind patterns are more zonally symmetrical. In January there is a poleward shift and /or slowing down of the jet stream when sea ice is low, which is strongest at lag -1 and weakest at lag +1. July and September exhibit an opposite pattern to January with an equatorward shift of the jet stream when sea ice is low, which is again strongest at lag -1 and weakest at lag +1, and also tends to strengthen with higher warming levels, particularly in September. The weakest and least significant relationships are found in March, particularly at lag +1.

## 5 Discussion

A direct comparison of our results with other studies is not straightforward due to differences such as experiment design, choice of model, coupling vs atmosphere only, sensitivities to model biases and background states and differences in amount and region of sea ice change. In particular, regression or composite based studies, such as this one, differ in their nature from modelling experiments where sea ice is perturbed, such as PAMIP. This is because the former capture two-way interactions between sea ice and atmospheric circulations, and the latter focus on the effects of sea ice on the atmosphere, making comparisons non-trivial. Lagged analysis overcomes this difference at least in part. Some overall features can nevertheless be discussed.

Firstly, concerning pan-Arctic sea ice variability, several aspects of the linkages we find with atmospheric conditions (specifically SLP and SAT) are supported by previous literature. For instance, warming over the area of sea ice reduction and cooling over the Northern midlatitude continents, and high pressure over Northern Eurasia, with pan-Arctic sea ice reduction has been found in several studies (e.g. Peings and Magnusdottir, 2014; Sun et al., 2015; England et al., 2018; Screen et al., 2018; Cohen et al., 2020 and references therein; Smith et al., 2022). Model sea-ice perturbation experiments tend to find a southward shift of the North Atlantic jet stream in response to pan-Arctic sea ice reduction. (Anderson et al., 2025; Barnes and Screen, 2015; Sun et al., 2015; Ye et al., 2023, 2024). A southward North Atlantic jet shift was also found for ACCESS-ESM1.5 in our composite analysis for some warming levels but in CESM2 it instead became less titled. Notably CESM2




showed little response of the North Atlantic jet in PAMIP simulations (Ye et al., 2023), despite a southward shift over the northern hemisphere as a whole in other PAMIP studies (Screen et al., 2022; Smith et al., 2022).

For regional sea ice variability, some key features of the associated SAT and SLP patterns found here also agree with those found in studies of selected regions using similar methods (regression or composite techniques), e.g. (Blackport and Screen, 2021; Cohen et al., 2020; and notably with Delhaye et al., 2024) who use the same region definitions). Not all regions have been equally well studied. One that has been the focus on several studies is the Barents-Kara Sea: Our analysis uncovered a delayed negative NAO response to low Barents-Kara sea ice, as has been found in many other studies of both types

(composite/regression and model perturbation) (Cohen et al., 2020; Delhaye et al., 2024; McKenna et al., 2018; Screen, 2017; Screen et al., 2018; Sun et al., 2015). A dependency of the pattern of remote cooling on the region of sea ice retreat has also been found in some sea ice perturbation experiments (Cohen et al., 2020; McKenna et al., 2018; Screen, 2017). A few such studies have also shown that the northern hemisphere jet on average, as well as the North Atlantic part, responds differently to sea ice reduction in different regions (Sun et al., 2015; Levine et al., 2021; Xu et al., 2024, Screen et a. 2017; McKenna et al.

2018). However, while model sea ice perturbation experiments also agree about there being differing responses to sea ice reduction in different regions, the details of these responses vary. In particular, SLP patterns differ between studies based on composite or regression analysis (Blackport and Screen, 2021; Delhaye et al., 2024) and model perturbation experiments e.g. (Levine et al., 2021; Screen, 2017). One exception is the abovementioned NAO response to Barents-Kara sea ice loss, a phenomenon that has been found in many model perturbation studies as well (Cohen et al., 2020 and references therein;

McKenna et al., 2018; Screen, 2017; Screen et al., 2018; Sun et al., 2015).

Concerning our lagged analysis, the evolution of the SAT pattern across lags seen here is also broadly similar to that found in other composite or regression studies, albeit with model differences in some of the details (Blackport and Screen, 2021; Delhaye et al., 2024; Kelleher and Screen, 2018). Moreover, such studies all agree that the relationships between sea ice and SLP are stronger when the atmosphere leads sea ice, than when sea ice leads the atmosphere, and that instantaneous

relationships seem to mostly reflect the SLP patterns contributing to sea ice reduction. They suggest, for example, that the high pressure over Eurasia, seen for the Pan-Arctic and Barents-Kara analyses here, drives sea ice decline in the Barents-Kara Seas by bringing in warm moist air, whilst also bringing cold air into Eurasia, and can also weaken the stratospheric polar vortex with subsequent downward impacts, i.e. a negative NAO. However, except for the Barents-Kara Sea ice reduction, they find low agreement between models on the SLP patterns following sea ice reduction (Delhaye et al., 2024).

For the Antarctic, there is less literature, and previous studies tend to be based on model experiments with perturbed sea ice, rather than a composite or regression analysis, and are therefore different in nature. However, model sea-ice perturbation experiments have found a negative SAM response to Antarctic sea ice retreat (Ayres and Screen, 2019; Bader et al., 2013) or a positive SAM for increased sea ice (Smith et al., 2017), similar to our composite analysis. Studies also tend to find an equatorward shift and slowdown in the jet in response to Antarctic sea ice reduction (Ayres et al., 2022; Ayres and Screen,

2019; Bader et al., 2013; England et al., 2018), or a poleward shift for an increase in Antarctic sea ice (Smith et al., 2017), although this can be somewhat season dependent (Ayres and Screen, 2019).





Finally, another key question behind our study was whether the relationships between sea ice variability and atmospheric circulation remain the same at different levels of background warming (and therefore different levels of background sea ice amount). While there has to our knowledge been no other investigations of polar to mid-latitude interactions for different

global warming levels, especially across several regions, existing studies suggest that there can be differences in the atmospheric responses for different amounts of sea ice decrease, even including a switch in sign of the NAO in some cases (Chen et al., 2016; McKenna et al., 2018; Peings and Magnusdottir, 2014; Petoukhov and Semenov, 2010; Semenov and Latif, 2015). However, (Chen et al., 2016) suggest that some of these differences could be due to internal variability due to an insufficient sample size or sensitivity to background state. We found that the atmospheric patterns associated with low Arctic

sea ice did evolve in some cases at higher warming levels, likely due to changing location of sea ice variability with global warming (i.e. at the retreating ice edge), whilst for the Antarctic some changes could be seen already across lower warming levels in some cases. We note that internal variability could be playing a role here, even with our large ensemble sizes, particularly at positive lags. Another factor could be differences in the amount of sea ice change between low and high states at different warming levels. However, larger changes in sea ice do not seem to correspond to larger magnitude circulation

patterns.

## 6 Summary

In this study we have investigated the linkages between interannual sea ice variability and atmospheric circulation using a composite analysis applied to two large ensembles of CMIP6 generation coupled climate model simulations (CESM2 and ACCESS-ESM1.5). We focused particularly on the dependency of these relationships on the sea-ice region examined for the

Arctic and on background levels of global warming (and therefore sea ice extent) for both hemispheres. We then used a lag analysis to try to disentangle the effect of the atmosphere on sea ice from the effects of the sea ice on the atmosphere.

One of our key questions was whether and how the relationships between sea ice variability and atmospheric circulation depend on the region of sea ice change. Our results show clearly that there is a strong regional dependency of the atmospheric circulation patterns associated with sea ice variations in different regions. For example, concerning instantaneous relationships,

low winter sea ice in the Barents-Kara Sea was accompanied with a positive NAO SLP pattern, with additional strong high pressure over northern Eurasia, while low Labrador sea-ice was associated with a negative AO pattern. For the Okhotsk, Chukchi, Bering and Greenland seas, a pressure dipole pattern with high pressure to the east and low pressure to the west of the region of low sea ice could be seen. SAT increased over and around the region of anomalously low sea ice with remote cooling elsewhere, for example, over midlatitude Eurasia for low sea ice in the Barents-Kara Seas, over North America for

low Chukchi and Bering sea ice, and over Northern Eurasia for low sea ice in the Labrador Sea. The area of maximum warming associated with low sea ice anomalies followed the retreat of the ice edge with increased global warming. The North Atlantic jet stream shifted southwards when sea ice was low in the Labrador Sea, shifted northwards or weakened for low sea ice in the Okhotsk Sea and strengthened and shifted northwards for low sea ice in the Chukchi-Bering Sea. These results were mostly



broadly similar between the two models examined, although with weaker and less widespread SLP patterns in ACCESS-ESM1-5 than CESM2.

This regional dependency exists both for instantaneous relationships and both positive and negative lags. Overall, the atmospheric patterns one or two months before low sea ice conditions (lag -1, -2) were very similar to the instantaneous patterns, suggesting that the latter mostly reflect the circulation patterns contributing to low regional sea ice. These results were largely consistent between the models. At positive lags, the SLP and jet patterns tended to be weaker and/or more different to the lag 0 patterns, less stable across global warming levels, and less consistent between models, with ACCESS-ESM1.5 often showing very weak patterns. Nevertheless, for low Barents-Kara and pan-Arctic sea ice, both models switched to a negative NAO at positive lags, consistent with previous studies. For SAT, again lag -2 and -1 already showed a similar pattern to lag 0, which faded away over subsequent months.

For the Antarctic we looked at various seasons. Again, SAT increased primarily over the regions with the largest low sea ice anomalies, shifting poleward with global warming and eventually spreading over the Antarctic continent in ACCESS-ESM1.5. The SLP patterns and southern hemisphere jet behaviour associated with low Antarctic sea ice were less consistent between the two models than for the Arctic. In austral winter and spring, a negative SAM pattern and equatorward shift of the polar jet were associated with low Antarctic sea ice in ACCESS-ESM1.5, and the opposite in Summer. In CESM2 these patterns were less zonally symmetric, but still consisted of high pressure over the pole, particularly in Autumn and Winter, whilst the jet strengthened and shifted equatorward over the Eastern South Pacific when sea ice was low in summer and autumn, and often slowed down elsewhere. Patterns were often strongest at lag -1, and weaker (ACCESS-ESM1.5) or somewhat evolved (CESM2) at lag +1 relative to lag 0. There was some degree of sensitivity of the patterns to warming level.

Another key question was whether or not the relationships between sea ice variability and atmospheric circulation remain the same at different levels of background warming (and therefore different levels of background sea ice amount). Overall, we found that atmospheric patterns associated with low sea ice were fairly consistent across warming levels, but did evolve in some cases at higher warming levels for the Arctic, and sometimes also at lower levels in the Antarctic.

Given the dependencies of the atmospheric circulation patterns on the region of sea ice variation in the Arctic, future studies are needed to explore whether Antarctic sea ice change in different regions has differing effects on the atmosphere. Concerning both poles, initiatives such as PAMIP (Smith et al., 2019) have been very helpful in overcoming difficulties in comparing results from different models due to differences in experimental set-ups. This includes experiments where sea ice is perturbed in the Barents-Kara and Okhotsk seas separately in atmosphere only models. However, it would be beneficial to examine the atmospheric responses to a greater number of sea ice regions in a multi-model context, particularly including coupled models since previous studies indicate that atmospheric responses tend to be stronger and more widespread with coupling (Ayres et al., 2022; Deser et al., 2015, 2016; England et al., 2020b, a).



## Data Availability Statement

The data from the ACCESS-ESM1.5 large ensemble are publicly available from the ESGF portals as part of CMIP6, for example https://esgf-data.dkrz.de/search/cmip6-dkrz/. The CESM2 large ensemble data is available from https://www.cesm.ucar.edu/community-projects/lens2/data-sets as cited in (Rodgers et al., 2021).

## Author contributions

All authors were involved in conceptualizing the study. CI performed the analysis. CI prepared the manuscript with contributions from all co-authors.

## Competing Interests

The authors declare that they have no conflict of interest.

## Acknowledgments

This research has received funding from the European Union's Horizon 2020 research and innovation programme under the CRiceS project, grant agreement no. 101003826. We acknowledge the World Climate Research Programme, which, through its Working Group on Coupled Modelling, coordinated and promoted CMIP6. We thank the climate modelling groups for producing and making available their model output, the Earth System Grid Federation (ESGF) for archiving the data and providing access, and the multiple funding agencies who support CMIP6 and ESGF. We also acknowledge the CESM2 Large Ensemble Community Project and supercomputing resources provided by the IBS Center for Climate Physics in South Korea.

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
