# Peer review of "How polar-to-midlatitude atmospheric teleconnections depend on regional sea ice fraction and global warming level"

_EGUsphere, 2025_

## Author Response (AR1)

**Responses to Reviewers**

We thank the reviewers and the editor for their valuable input and suggestions on our manuscript. Responses to the individual points are given inline below. The main changes are as follows:

- We edit the manuscript to make it clearer what the goals of the study are and are not: i.e. we do not aim to conduct a detailed analysis of cause and effect regarding connections between sea ice and atmospheric circulation, but rather we build on such studies that do focus on this to focus instead on whether and how these connections evolve with the amount of background warming, which is a new angle. This is made possible by the use of large ensembles that allow a large sample of internal variability across a range of warming levels.
- We test the sensitivity of the composite analysis to using different percentile thresholds to define what counts as low and high sea ice states.
- We add some material from the literature concerning how well the two climate models represent modes of variability and atmospheric circulation features of the extratropics.

**Reviewer 1**

Summary

This manuscript investigates how atmospheric teleconnections between polar and mid-latitude regions depend on regional sea ice variations and global warming levels. Using two large ensemble climate model simulations (CESM2 and ACCESS-ESM1.5), the authors employ various statistical analysis to examine relationships between sea ice variability and atmospheric circulation patterns (surface air temperature, sea level pressure, and jet streams) across different Arctic regions and Antarctic conditions. They found strong regional dependency and persistence of teleconnection patterns till 3~4C warming.

*We thank the reviewer for taking the time to review our manuscript and for the helpful comments and suggestions.*

Major comments

1. The composite analysis - major analysis method used - cannot cleanly separate cause and effect, as acknowledged by the authors. While lagged analysis is attempted, the high autocorrelation in sea ice makes interpretation challenging.

*We agree with the reviewer that we cannot cleanly separate cause and effect using the composite technique, but we emphasize that disentangling cause and effect is not the main goal of the paper. Rather, we build on previous studies that focus more on cause and effect to explore a new angle: the extent to which atmospheric teleconnections between sea ice and the midlatitudes persist or change with different levels of background global warming. Thus, we only aim to get a first order idea of what is cause and effect and to verify that these findings are consistent with previous studies. Instead, we focus on the novel parts: which is primarily the warming level analysis, and further applying it to a range of regions. This is made possible by the*

*use of large ensembles that allow a large sample of internal variability across a range of warming levels.*

*We edit the text in various places to make it clearer that this is only a first order look at cause and effect:*

*Line 100-102 where the lagged analysis is introduced in the introduction "we also perform a lagged analysis as a first order attempt at disentangling the two. We emphasize, however, that a detailed examination of causality and mechanisms is not the focus of this study, but rather we build on the findings of previous more process-oriented studies to focus on whether these teleconnection patterns persist across different levels of background global warming."*

*Line 199: "…, and then examine associations when sea ice or the atmosphere lead or lag by a couple of months, to get a first order idea of what is cause and what is effect."*

*We also add the following to the lagged analysis section starting at line 295 "However, we emphasize that a thorough exploration of cause and effect is not the main focus of the paper, and we refer to previous studies for more detailed process understanding. Rather, we focus on how these teleconnections evolve with global warming and how they depend on the region of sea ice variability."*

*Line 539 : "We then used a lag analysis as a first order attempt at disentangling the effect of the atmosphere on sea ice from the effects of the sea ice on the atmosphere."*

2. The 30th/70th percentile thresholds are too arbitrary. If the other values are used, then the result changes?

*We repeat the results for figures 4 and 6 (i.e. the sea level pressure composites and the lagged version) using $20^{th}/80^{th}$ and $40^{th}/60^{th}$ percentiles. The results are consistent and are included as an attachment. We add the following to the text at line 146: "(Results were insensitive to using a stricter (20th and 80th percentiles) or more lenient (40th and 60th percentiles) threshold, tested for the SLP composites for CESM2 (not shown))."*

3. Model validation: I could not find much of information about how well two models perform in terms of teleconnection.

*We thank the reviewer for this suggestion, and add some information on this based on previous literature at lines 124-137:*

*"In terms of model performance for the main extratropical atmospheric circulation features, we base the following on evaluations in Simpson et al., (2020) and Coburn and Pryor, (2021). The southern hemisphere jet stream is well simulated in terms of its latitude in both models, although it is a little equatorward biased in ACCESS-ESM1.5 in austral winter. Nevertheless, it is too strong, particularly in CESM2. The winter North Atlantic jet stream is well simulated in terms of its latitude and tilt in CESM2, but is also too strong. In ACCESS-ESM1.5 the tilt is similar to CESM2 and the winds are weaker (e.g. contrast contours in Figure 7 and S11). In terms of the main modes of variability, both models capture the overall spatial pattern of the Southern Annular mode well, although it is a bit too zonally symmetric, particularly in ACCESS-ESM1.5, a bias common in many CMIP6 models. In terms of the northern annular mode, both models*

*capture the overall spatial structure, although the pressure center over the North Pacific and to a lesser extent over Northern Russia is too strong in both models, particularly in CESM2, although these are again common model biases amongst CMIP6 models. The structure of the winter NAO is well simulated in CESM2 and the three main centers of northern hemisphere winter blocking are captured, although with low biases over Europe and Greenland. However southern hemisphere blocking is substantially underestimated. These underestimates are mostly due to mean state biases. Equivalent information for the NAO and blocking was not available for ACCESS-ESM1.5.*

4. Antarctic result: The Antarctic analysis feels somewhat tacked on, with less thorough investigation than the Arctic, e.g,, no regional analysis.

*While we have no regional analysis for the Antarctic, we do instead include seasons. The reasons for not including regions for the Antarctic is partly because the paper would become too long if we did so, but also there is a lack of previous literature on responses of the climate to regional variations of Antarctic sea ice with which we can compare our findings before moving on to focus on the sensitivity to global warming level.*

5. This is somewhat related to #1 and #3. Monthly teleconnection between polar and mid-latitude has quite a bit of debate. There are quite a bit of work on daily dataset to tackle on teleconnection and its future (e.g., Kug et al. 2015, Wu et al. 2022, Hong et al. 2024). It is okay to focus on monthly time scales. However, issues like existence of such teleconnection, model validation, and causality need to discussed.

*We thank the reviewer for this suggestion and the references. However, despite our best efforts we were unable to find out which papers these were without the full references. However, for the composite technique that we use here, we feel that using monthly data should give a better signal to noise ratio than doing the composites on a daily level, and is less confusing when it comes to looking at lags. Other studies based on composite or regression techniques also use monthly or seasonal data* (Blackport and Screen, 2021; Delhaye et al., 2024; Kelleher and Screen, 2018).

*We also add some text on model validation as described in the response to comment 3. Please see the response to comment 1 regarding causality.*

6. While comprehensive as the target, many individual findings confirm previous studies. I really have to ask what are the key findings here?

*The main novelty of the study is to explore how these teleconnections evolve or otherwise with level of background global warming and to do this across multiple sea-ice regions. A systematic exploration of these linkages across global warming levels has not been done before, particularly across multiple regions.*

Overall, this manuscript has really wide range of the scope, which is good. However, there are numerous issues. For example, one could ask perform the Granger causality analysis in response to #1. Or, sensitivity for those threshold and detailed model validation could be suggested. Then, the analysis could be even more comprehensive without clear focus.

*Granger causality analysis applies to time series rather than to composite methods, and so is not applicable here. In addition, we emphasize that disentangling cause and effect is not the main goal of this study. Rather we build on other studies that have explored cause and effect to explore a new angle: the extent to which linkages between sea ice and atmospheric teleconnections persist at different levels of background warming. We test the sensitivity of the results to percentile threshold of sea ice amount as described in response to comment 2 and we add some description of model performance in terms of jet streams and modes of extratropical variability in response to comment 3.*

**Review for "How polar-to-midlatitude atmospheric teleconnections depend on regional sea ice fraction and global warming level" by Iles, Samset and Lund**

Summary

In this manuscript, the authors use composite analysis with two climate model large ensembles to investigate the teleconnections associated with Arctic and Antarctic sea ice interannual variability and how these relationships evolve under different levels of global warming. The lagged composite analysis shows that the instantaneous relationships mainly reflect atmospheric conditions contributing to low sea ice, while the patterns become weaker or altered when sea ice leads. The authors also demonstrate that the teleconnections are generally robust across warming levels.

Overall, the topic fits well within the scope of Earth System Dynamics, and the authors provide a solid review of relevant literature. However, I have fundamental concerns regarding the methodology, particularly the use of instantaneous composites, and the interpretation of results derived from this approach. I would be happy to recommend publication once these issues are satisfactorily addressed. I hope my comments below will help strengthen the manuscript.

*We thank the reviewer for their careful review of our manuscript and for their helpful comments.*

Major comments:

1. The reliance on instantaneous composites does not provide sufficient scientific insight. The authors build on Delhaye et al. (2024), who performed a similar analysis using CMIP6 preindustrial simulations. However, that study explicitly examined both the precursors of regional sea ice loss and the potential atmospheric impacts of sea ice anomalies, particularly in the Barents–Kara Seas. In contrast, the present manuscript emphasizes instantaneous composites, which by themselves lack a clear physical interpretation because they combine both forcing and response signals. As a result, much of the analysis describes patterns without clarifying the underlying mechanisms. This represents a key weakness of the study and could be considered a major limitation.

Related to this point, it is difficult to reconcile the results with much of the previous literature, which focuses on the atmospheric response to sea ice forcing. For example, the discussion section is problematic because the identified teleconnections appear to reflect circulation

anomalies that precede sea ice retreat, yet they are compared to studies that investigate the atmospheric response to imposed sea ice loss.

To address this issue, I encourage the authors to place greater emphasis on the lagged analysis, similar to Delhaye et al. (2024). Doing so could help separate atmospheric precursors of sea ice variability (e.g., lag –1 or 0) from the potential impacts of sea ice anomalies on the atmosphere (e.g., lag +1). Such a framing would provide stronger physical interpretation and closer alignment with prior work.

*We thank the reviewer for these comments. In the revision, we have made it clearer that the main point of our study is not to repeat earlier studies looking into the mechanisms linking atmospheric patterns and sea ice changes, but rather to build on them in order to (1) investigate whether similar results are recovered using the alternative pattern identification method employed here for large ensembles, and (2) investigate whether the patterns persist for higher levels of global warming. While it can be expected that the patterns change when sea ice is more or less gone from a given region, as we indeed find, it is not apriori given that they are independent of the overall shifts in global atmospheric circulation that occurs with global warming. That we still added sections on lead/lag relations was because we, like the reviewer, are interested in the causal relations too, and would like to further establish whether they are consistent between our method and other studies – notably Delhaye et al. 2024.*

*In the revision, we have clarified the goals of the study, and added some text to the introduction in line with the above explanation that we build on established patterns rather than trying to reproduce them. We also make edits in several places to emphasize that the lagged analysis is only a first order attempt at disentangling cause and effect (see response to reviewer 1 comment 1) and change the wording in a number of places where we had unintentionally implied a direction of causality (e.g. in the title and abstract, but also elsewhere).*

*For the second point made above, regarding comparisons to studies that have explicitly imposed a sea ice change, we have revised the discussion section and made clear which lags from our study are comparable to other studies, as well as what their setup was. Hopefully this helps in clarifying where the present study sits in the broader literature on the relationships between atmospheric circulation and sea ice changes.*

*We also replace the word "instantaneous" with "concurrent" to describe the lag 0 composites throughout as we felt the word "instantaneous" could be misleading e.g. for anyone working on climate forcing for whom instantaneous means literally instantly (at the speed of light).*

2. While the lag analysis helps distinguish circulation patterns that force sea ice variability, it remains difficult to fully separate forcing and response signals. I have two suggestions that might help strengthen the analysis. While I cannot guarantee they will resolve the issue, they may provide additional insight into the two-way interactions:

- The authors emphasize the importance of coupled model simulations for capturing the full range of sea ice–atmosphere–ocean interactions. However, atmosphere-only simulations can still provide valuable perspective on the atmospheric response to sea

ice anomalies. For example, CESM2 offers an 11-member atmosphere-only ensemble over 1950–2014 as part of CMIP6, and an additional 10-member ensemble covering 1880–2019 is available at https://gdex.ucar.edu/datasets/d651010/. Since the teleconnections appear robust across global warming levels, these simulations may be used to isolate the atmospheric response to polar sea ice loss, at least for one GSAT level.

- It may be possible to more clearly separate circulation patterns that precede sea ice retreat (e.g., lag –1) from those that respond to sea ice anomalies (e.g., lag +1). Yook et al. (2012) applied a congruence method to distinguish forcing and response patterns in the context of Kuroshio–Oyashio variability, and a similar approach could potentially be adapted here.

*We thank the reviewer for these suggestions also, which would indeed allow us to investigate the underlying mechanisms more clearly. However, as stated above, the point of the present study is to build from existing mechanistic understanding and exploit the information available from large ensembles to look at something that other datasets can't. Hence, we have chosen this particular method for the present study. For a more mechanistically oriented follow-up, however, these points would be of great interest. (We also note that sea ice data does not seem to be available for the second suggested ensemble, although there are 7 simulations with sea ice available from AMIP.  This could be supplemented by running a new set of simulations. It is however not possible for us to do this for the present study.) We do however add these suggestions to the end of the conclusion section as ways in which this work could be extended.*

3. The manuscript currently contains an excessive number of figures, with many panels (often 20–40 per figure). However, most of these figures do not substantially advance the physical understanding of the processes in question, particularly regarding the forcing versus response of sea ice loss. I encourage the authors to carefully consider whether it is necessary to show all five lags (–2, –1, 0, +1, +2) across all five warming levels (0–4 °C). Streamlining the presentation would make the paper more accessible and focused. Moreover, Figures 7 and 10 do not appear to add significant value in their current form, unless they can be more directly linked to explaining the mechanisms of regional sea ice loss in the Arctic and Antarctic.

*The physical understanding gained by these figures is, for the most part, the evolution with global warming level. See also our first response above. This is why we include a broad set of warming levels and lags, to let the reader see this evolution visually. Note that in most cases the figures are already showing a subselection of warming levels and lags, to make them manageable. Looking at these teleconnections across different warming levels is the focus of this paper, so it is necessary to show a certain number of warming levels. We do, however, remove Figures S11 and S9, which were similar to Figure 6 and S10, but for lag -2 and +2 rather than -1 and +1.*

*Jet streams are included as an example of a downstream impact relating to the teleconnections and circulation patterns, rather than as an explanatory factor in itself. Northen European storms, and hence extreme events, are strongly related to the North Atlantic storm track, which are, in*

*turn, related to the jet stream position. We therefore show how this pattern evolves with global warming as function of the amount of sea ice. In a hypothetical situation where we have a well validated understanding of this relation, the results of this section could be used as an early warning for the location of European storms as function of the amount of Arctic sea ice – or for sea ice loss, depending on the direction of causality. As of now, it serves as an indicator of how sensitive the relationship between storm track location and amount of sea ice is in different Arctic regions, as function of global warming, in the particular models we've used.*

4. Instead of constructing composites based on GSAT warming levels, an alternative would be to first remove the ensemble-mean forcing signal and then repeat the analysis over moving 50-year windows across the historical and SSP periods. I am not suggesting that the current approach is wrong, but I wonder if there is a specific reason the authors chose to base the analysis on GSAT. One potential limitation is that GSAT can be influenced by internal variability, such as the Interdecadal Pacific Oscillation (IPO) and Atlantic Multidecadal Variability (AMV), which may complicate the interpretation of GSAT-based composites.

*We thank the reviewer for the suggestion. We could indeed have performed the analysis differently, but chose the global warming levels as this is a well-tested method for investigating the overall evolution of climate features with increasing surface warming. It was used extensively in the IPCC AR6 for instance (see e.g. Chapter 11, where the methodology is explained in a box, although we use a yearly definition rather than a 20-year period). We also used and tested various versions of this method in this paper: Samset et al. 2019; https://agupubs.onlinelibrary.wiley.com/doi/full/10.1029/2019EF001160. It is true that IPO, AMV, ENSO and similar modes will be part of the composites. However, the method is designed (and tested) to be robust to this. As an example, the ENSO variability shifts global mean temperature up and down by up to 0.5 °C, in effect shifting years into or out of a given "level" of global warming. Since we use years on both sides of the diagnosed level of warming, as well as include years from multiple ensemble members, this contribution however averages out in the final figures. The same will be true for e.g. AMV which has a more direct influence on sea ice levels, as any bias introduced would come via its influence on global mean temperature. The above paper also includes some tests concerning these statements, as does the IPCC AR6 Chapter 11 and its assessed literature. See also the plot below, where we show which years from the full ensemble are included in the various warming levels for GSAT itself and for Pan-Arctic sea ice area.*

[Figure]

Figure R5: January Pan-Arctic sea ice area from CESM2 showing the years selected for each global warming level (relative to preindustrial 1850-1900).

[Figure]

Figure R6: Annual mean global surface air temperature from CESM2 showing the years selected for each warming level.

5. Have the authors examined surface energy flux anomalies in the lagged analysis? In principle, the forcing and response phases should show opposite-signed anomalies (i.e., downward fluxes for the forcing circulation, upward fluxes for the atmospheric response). Including such an analysis could provide additional physical insight and help distinguish between precursor and response signals.

*We have not done this for the present paper because, as stated above, establishing causality was not our major aim. It would be possible to perform such an analysis using large ensembles, however it would require extensive validation of the modelled surface energy flux for known perturbations before clear conclusions could be drawn. This is unfortunately not within the scope of the present analysis, although, as for the previous suggestions, it is a good opportunity for a follow-up study directed more specifically towards disentangling mechanistic causes and effects. We add this to the suggestions for further work at the end of the conclusions section.*

Reference:

Yook, S., D. W. J. Thompson, L. Sun, and C. Patrizio, 2022: The Simulated Atmospheric Response to Western North Pacific Sea Surface Temperature Anomalies. J. Climate, 35, 3335–3352, https://doi.org/10.1175/JCLI-D-21-0371.1.

**References**

*Blackport, R. and Screen, J. A.: Observed Statistical Connections Overestimate the Causal Effects of Arctic Sea Ice Changes on Midlatitude Winter Climate, J Clim, 34, 3021–3038, https://doi.org/10.1175/JCLI-D-20-0293.1, 2021.*

*Delhaye, S., Massonnet, F., Fichefet, T., Msadek, R., Terray, L., and Screen, J.: Dominant role of early winter Barents–Kara sea ice extent anomalies in subsequent atmospheric circulation changes in CMIP6 models, Clim Dyn, 62, 2755–2778, https://doi.org/10.1007/s00382-023-06904-6, 2024.*

*Kelleher, M. and Screen, J.: Atmospheric precursors of and response to anomalous Arctic sea ice in CMIP5 models, Adv Atmos Sci, 35, 27–37, https://doi.org/10.1007/s00376-017-7039-9, 2018.*

**20-80th percentiles. Sea level pressure for low minus high sea ice, January, CESM2**

Figure R1: As for Figure 4 but using thresholds of the 20th and 80th percentiles to define low and high sea ice amounts respectively (instead of the 30th and 70th percentiles as in Figure 4).

**40-60th percentiles. Sea level pressure for low minus high sea ice, January, CESM2**

Figure R2: As for Figure 4 and R1 but using thresholds of the 40th and 60th percentiles to define low and high sea ice amounts respectively.

**20-80ᵗʰ percentiles. Lagged SLP for low minus high January sea ice, CESM2**

[Figure]

Figure R3: As for Figure 6 but using the 20ᵗʰ and 80ᵗʰ percentiles to define low and high sea ice states respectively (instead of the 30ᵗʰ and 70ᵗʰ percentiles as in Figure 6).

[Figure]

Figure R4: As for Figure 6 and R3 but using the 40th and 60th percentiles to define low and high sea ice states respectively.

---

## Author Response (AR2)

**EGUSPHERE-2025-4115**

**Authors Responses**

How polar-midlatitude atmospheric teleconnections depend on regional sea ice fraction and global warming level

Carley Elizabeth Iles, Bjørn Hallvard Samset, and Marianne Tronstad Lund

**Editor comment**

The authors have done an excellent job responding to the review comments. Reviewer #1 has a minor suggestion that I would like the authors to consider. Other than that, great work!

*Thank you for your encouraging words! Please see the response to reviewer 1's comments below.*

**Reviewer 1**

The authors have revised the manuscript in response to previous reviews, clarifying scope, improving framing, adding model-skill context, and testing methodological sensitivity. The revised manuscript reads more clearly and the scientific purpose is now better articulated. Yet, I just have one minor suggestion.

Antarctic section: The lack of regional decomposition is somewhat understandable, but the text could more explicitly explain why Antarctic treatment is intentionally streamlined and what future work could expand.

*We now add the following sentence to the start of the Antarctic section (section 4, line 395-398): "We leave an analysis of regional dependencies of Antarctic sea ice- atmospheric circulation linkages for future work, partly due to space constraints, but also due to a lack of previous literature on atmospheric responses to regional Antarctic sea ice variations with which to compare our findings before moving on to explore dependencies on global warming level."*

*We thank the reviewer for taking the time to review our manuscript and for their helpful suggestions.*

Reviewer 2

I appreciate the authors' responses and revisions. While I would have preferred that the authors further pursue the causal interpretation in this study, I understand their decision to leave this to future work. Aside from this limitation, I see no reason to prevent publication.

*We thank the reviewer for taking the time to review our manuscript and for their understanding regarding the causality aspect.*